# Impact of SARS-CoV-2 Infection on Humoral and Cellular Immunity in a Cohort of Vaccinated Solid Organ Transplant Recipients

**DOI:** 10.3390/vaccines11121845

**Published:** 2023-12-13

**Authors:** Bernardo Ayala-Borges, Miguel Escobedo, Natalia Egri, Sabina Herrera, Marta Crespo, Sonia Mirabet, Carlos Arias-Cabrales, Anna Vilella, Eduard Palou, María M. Mosquera, Mariona Pascal, Jordi Colmenero, Marta Farrero, Marta Bodro

**Affiliations:** 1Unit for Heart Failure and Heart Transplantation, Department of Cardiology, Hospital Clínic, L’Institut d’Investigacions Biomèdiques August Pi i Sunyer (IDIBAPS), Universitat de Barcelona, 08036 Barcelona, Spain; ayala@clinic.cat; 2Liver Transplantation, Liver Unit, Hospital Clínic, L’Institut d’Investigacions Biomèdiques August Pi i Sunyer (IDIBAPS), University of Barcelona, 08036 Barcelona, Spain; escobedo.miguel.b@gmail.com (M.E.); jcolme@clinic.cat (J.C.); 3Department of Immunology, Hospital Clínic, L’Institut d’Investigacions Biomèdiques August Pi i Sunyer (IDIBAPS), University of Barcelona, 08036 Barcelona, Spain; egri@clinic.cat (N.E.); epalou@clinic.cat (E.P.); mpascal@clinic.cat (M.P.); 4Department of Infectious Diseases, Hospital Clínic, L’Institut d’Investigacions Biomèdiques August Pi i Sunyer (IDIBAPS), University of Barcelona, 08036 Barcelona, Spain; sherrera@clinic.cat; 5Centro de Investigación Biomédica en Red de Enfermedades Infecciosas (CIBERINFEC), Instituto de Salud Carlos III, 28029 Madrid, Spain; 6Nephrology Department, Renal Transplant Unit, Hospital del Mar Research Institute, Hospital del Mar, 08003 Barcelona, Spain; mcrespo@psmar.cat (M.C.); cariascabrales@psmar.cat (C.A.-C.); 7Heart Transplantation Unit, Department of Cardiology, Hospital Sant Pau, Centro de Investigación Biomédica en Red Enfermedades Cardiovaculares (CIBERCV), 08041 Barcelona, Spain; smirabet@santpau.cat; 8Department of Preventive Medicine and Epidemiology, Hospital Clínic, L’Institut d’Investigacions Biomèdiques August Pi i Sunyer (IDIBAPS), Universitat de Barcelona, 08036 Barcelona, Spain; avilella@clinic.cat; 9Microbiology Department, Hospital Clínic de Barcelona, Institute for Global Health (ISGlobal), University of Barcelona, 08036 Barcelona, Spain; mdmosquera@clinic.cat; 10Department of Medicine, Faculty of Medicine, University of Barcelona, 08036 Barcelona, Spain; 11Department of Infectious Diseases, Hospital Clínic of Barcelona, Carrer Villarroel, 08036 Barcelona, Spain

**Keywords:** COVID-19, solid organ transplant recipients, vaccination, humoral immunity, cellular, immunity

## Abstract

The aim of the present study was to determine humoral and T-cell responses after four doses of mRNA-1273 vaccine in solid organ transplant (SOT) recipients, and to study predictors of immunogenicity, including the role of previous SARS-CoV-2 infection in immunity. Secondarily, safety was also assessed. Liver, heart, and kidney transplant recipients eligible for SARS-CoV-2 vaccination from three different institutions in Barcelona, Spain were included. IgM/IgG antibodies and T cell ELISpot against the S protein four weeks after receiving four consecutive booster doses of the vaccine were analyzed. One hundred and forty-three SOT recipients were included (41% liver, 38% heart, and 21% kidney). The median time from transplantation to vaccination was 6.6 years (SD 7.4). In total, 93% of the patients developed SARS-CoV-2 IgM/IgG antibodies and 94% S-ELISpot positivity. In total, 97% of recipients developed either humoral or cellular response (100% of liver recipients, 95% of heart recipients, and 88% of kidney recipients). Hypogammaglobulinemia was associated with the absence of SARS-CoV-2 IgG/IgM antibodies and S-ELISpot reactivity after vaccination, whereas past symptomatic SARS-CoV-2 infection was associated with SARS-CoV-2 IgG/IgM antibodies and S-ELISpot reactivity. Local and systemic side effects were generally mild or moderate, and no recipients experienced the development of de novo DSA or graft dysfunction following vaccination.

## 1. Introduction

Vaccination has been one of the most effective tools in the fight against the COVID-19 pandemic, as demonstrated in clinical trials [1,2]. However, solid organ transplant (SOT) recipients have not been represented in these studies. Therefore, the evidence for this group is scarce. In addition, SOT recipients are among the most vulnerable groups, presenting higher morbidity and mortality due to COVID-19 infection [3]. Assessing the risk–benefit trade-off, the European and American transplant associations have recommended vaccinating these patients [4,5].

SOT recipients have a depressed antibody and cell immune responses, resulting in lower protection with conventional vaccination schemes [6]. A complete immune response, involving antibodies and cells, to the mRNA-1273 vaccine for SARS-CoV-2 was assessed by our group in a previous study that only included liver and heart recipients, evidencing that 41% developed humoral immunity after the first dose and 57% after the second dose [7]. Other studies with the two-dose schemes have also shown a lower immune response in SOT recipients compared to the general population [8,9].

To overcome the poor immune response in SOT recipients, the application of boosters has been proposed to achieve adequate levels of anti-SARS-CoV-2 antibodies. Recent studies have shown an increasing immunological response with the successive administration of a fourth and fifth dose, reaching up to 94% seroconversion with the fifth dose [10,11].

Most studies have focused on the humoral response by assessing SARS-CoV-2 IgG antibody levels [12]. However, the cellular response significantly contributes to preventing SARS-CoV-2 infection and the development of severe disease. Rezahosseini et al. evaluated the cellular immune response to at least three doses of mRNA vaccine in SOT recipients, finding that 46% of the SOT recipients developed a cellular response, compared to 69% in the control group [13,14].

The aim of this manuscript was to assess immune response (cell and antibody) to four doses of mRNA-1273 vaccine in a cohort of liver, heart and kidney transplant patients. The secondary objectives were to evaluate the side-effects of the mRNA-1273 vaccine, to ascertain the development of breakthrough infection and its severity, and to establish predictors of immune response among vaccinated SOT recipients.

## 2. Materials and Methods

Liver, heart, and kidney transplant recipients from three different institutions in Barcelona, Spain, received four doses of the mRNA-1273 SARS-CoV-2 vaccine. The first dose of 100 mcg was administered in the deltoid area, followed by the second dose four weeks later. The third and fourth doses were administered almost six months after the previous dose, following our national policy. Patients who received multi-organ transplants were excluded from the study.

Patients who provided their consent to participate in the study signed the informed consent forms, and we collected blood samples at two time points: baseline (one hour before each vaccine dose) and after 4–6 weeks of each dose, following the protocol established in our previous study [7]. The primary endpoint encompassed assessments at all time points, including the evaluation of humoral responses (IgM/IgG) against the spike (S) protein and the examination of cellular responses to the SARS-CoV-2 virus S protein. The secondary endpoint was to assess characteristics related to a lack of vaccine-induced response (antibody, cell or both). To ensure the recording of adverse events related to the vaccine, phone inquiries were performed 1–2 days after each dose. A pre-set survey was used to assess local and systemic symptoms, categorizing their severity on a semiquantitative scale (none/mild/moderate/severe).

The Institutional Ethics Committee accepted the study (HCB/2021/0222).

### 2.1. Patient Information

Patient information was compiled for examination: age, sex, body mass index (BMI), coexisting conditions, transplant date, immunosuppressive therapy details, and the utilization of immunosupressive therapy within the past year. Additionally, we gathered information on lymphopenia (lymphocyte count ≤ 1000/mm^3^), hypogammaglobulinemia (≤6.8 g/L of total IgG upon enrollment), ultrasensitive troponin levels, liver function tests, and chronic kidney disease (CKD) (defined as maintaining a glomerular filtration rate (GFR) below 60 mL/min/1.73 m^2^ in the last 3 months).

Organ function parameters such as Troponin I and transaminases were analyzed at three distinct time points: at baseline, 2 weeks before and 4 weeks after the second vaccine shot. Donor-specific HLA antibodies were analyzed at baseline and a month after the second vaccine shot using a single bead assay on a Luminex platform. Screening was performed with the Lifecodes LifeScreen Deluxe kit (Lifecodes, a division of Immucor, Stamford, CT, USA), and any beads surpassing an MFI (mean fluorescent intensity) of 3000 were deemed positive.

SARS-CoV-2 infection was based on hospital admission in patients that needed hospitalization or self-reporting (including a confirmatory test). All participants were called periodically (every 6 months) to assess symptomatic infection.

### 2.2. Detection of Antibodies to SARS-CoV-2

During the initial phase (from the first to the third vaccine dose), we quantified immunoglobulin titers using a serological assay based on the Luminex technique, known for its broader dynamic range compared to other methods. Antibodies directed against the receptor-binding domain of the S protein via Luminex were measured.

During the second phase of the study (after the third vaccine dose), IgG and IgM antibodies specific to SARS-CoV-2 were assessed in serum using a chemiluminescent immunoassay (CLIA) performed with an Atellica^®^ IM analyzer (Siemens Healthcare GmbH). The unit of measurement used, Binding Antibody Units (BAU), is in accordance with the latest guidance from the World Health Organization (WHO).

### 2.3. Detection of Cellular Response by IFN-γ ELISpot

To perform the ELISpot test, we used 2 × 10^5^ PBMCs, which were supplemented in X-VIVO^TM^ 15 medium (LonzaBasel, Switzerland) with 10% heat-inactivated AB serum and PepTivator^®^ SARS-CoV-2 Prot_S peptide pools* (1 µg/mL, Miltenyi Biotec, Bergisch Gladbach, Germany). Spot-Forming Units (SFU)/2 × 10^5^ PBMCs were used to quantify the results. A cut-off of >6 SFU/2 × 10^5^ PBMCs at ±2 SD of SFU/2 × 10^5^ PBMCs was established (healthy donors obtained before the SARS-CoV-2 pandemic).

In patients showing a non-reactive test, an additional human TNF-α/IL-2 double-color Enzymatic ELISpot assay (ImmunoSpot, Osaka, Japan) was performed according to the manufacturer’s guidelines to detect T cell responses. SARS-CoV-2–specific spots of TNF-α/IL-2 and IL-2 were determined by spot increment, defined as stimulated spot numbers ≥ 6 SFU/2 × 10^5^ PBMC.

### 2.4. Detection of Anti-HLA Donor-Specific HLA Antibodies

An additional assessment of donor-specific HLA antibodies both at the initial time point and four weeks following the second dose was conducted, employing the Luminex-based bead assay technique to perform this screening. The samples were analyzed with the Lifecodes LifeScreen Deluxe kit (Lifecodes, Immucor, Stamford, CT, USA). When the screening yielded positive results, we determined the specific HLA antibody types using the same assay. An MFI greater than 3000 was considered a positive result.

### 2.5. Statistical Analysis

Continuous variables were characterized with either the mean and standard deviation or the median and interquartile range, according to their characteristics. Categorical variables were described in terms of absolute frequencies or percentages. To explore the association between clinical status and vaccine unresponsiveness, univariable logistic regression was employed. Variables that showed association with the primary outcome at a significance level of *p* < 0.1 were subsequently included in the multivariable analysis. Variations in ELISpot response and antibody levels at follow-up were assessed using the Wilcoxon test for paired samples. Differences in ELISpot forming units and antibody titers between groups were analyzed using the Mann–Whitney test. All statistical tests were carried out using a confidence interval of 95%, and a *p*-value < 0.05 was considered statistically significant. The software SPSS v.25 (SPSS Inc., Chicago, IL, USA) was used to perform the analysis. Figures were created with GraphPad v.5 (GraphPad Software, La Jolla, CA, USA).

## 3. Results

### 3.1. Baseline Characteristics

In total, 143 SOT recipients were included: 59 (41%) were liver transplant recipients (LTR), 54 (38%) were heart transplant recipients (HTR), and 30 (21%) were kidney transplant recipients (KTR). Figure 1 shows the study design and the number of included patients at different time points of the study. Table 1 summarizes baseline characteristics. The median age was 61 years, with 35% of the total population being women. Dyslipidemia and hypertension were more frequent in KTRs (86.7% and 100%, respectively). LTRs (25%) were more commonly vaccinated during the first year after transplant than KTRs (10%) or HTRs (7.4%). Triple or quadruple immunosuppressive therapy was more common in HTRs (80%) than in KTRs (63%) or LTRs (17%). HTRs and KTRs were more frequently treated with prednisone and mycophenolate compared to LTRs. HTRs received higher doses of mycophenolate than other recipients. Acute graft rejection in the previous year was more frequent in LTRs.

### 3.2. Antobody and Cell Response to the mRNA-1273 Vaccine

Figure 2 illustrates the evolution of the antibody (anti-SARS-CoV-2 antibodies) and cellular (ELISpot) responses to successive doses of the mRNA-1273 vaccine up to the fourth dose. The humoral response shows a progressive increase in response rates until reaching a plateau of approximately 90%. Conversely, the cellular response shows a decrease in response rate between the second and third doses. However, after the third dose, there is evidence of a progressive increase in the response rate, reaching 94% of response after the fourth dose.

Comparing to immunized patients (after the second dose of vaccine), spots for the S protein significantly increased from 9 [0–147] to 13 [0–469] (*p* < 0.001) after the third dose and 15 [0–2016] after the fourth dose (*p* < 0.001).

Anti-S SARS-CoV-2-specific IgG increased from 1.8 [0.05–10] (after the first dose of vaccine) to 4.32 [0.05–10] (after second dose) (*p* < 0.001), and from 4 [0.1–15.6] before the third dose to 8 [0.1–14.7] after third dose (*p* < 0.001). To assess antibody response before and after the fourth vaccine dose, we used a chemiluminescent immunoassay and found a mean value of 1804 [0.6–3270] before the fourth dose and 2947 [0.6–5680] after the fourth dose (BAU/mL).

Based on the type of transplanted organ, liver recipients presented higher Anti-S SARS-CoV-2-specific IgG antibodies after second dose (7.2 [0.1–10]) compared to kidney (2.3 [0.1–10]) and heart recipients (0.6 [0.05–6.2]), *p* < 0.001. After the third and fourth doses, there were no statistically significant differences in the mean values between different organ recipients.

The mean spot-forming units (SFU) in liver recipients were 24 (0–44) after the second vaccine dose, 29 (0–469) after the third vaccine dose and 28 (0–216) after the fourth vaccine dose. In heart recipients, the mean SFU values were 10 (0–32) after the second vaccine dose, 12 (0–90) after the third and 21 (0–124) after the fourth vaccine dose. Finally, kidney transplanted patients presented 15 (0–52) SFU after the second dose, 12 (0–88) after the third, and 15 (1–58) after the fourth vaccine dose. Spots for the S protein were statistically different (*p* < 0.001).

### 3.3. Factors with Association to Vaccine Unresponsiveness

Ninety-three percent of SOT patients developed humoral immunity after four doses of the mRNA-1273 vaccine. Table 2 displays the results of the logistic regression analysis examining the factors associated with a lack of humoral response (IgG antibodies absence) six weeks after the fourth dose of the mRNA vaccine. Hypogammaglobulinemia was the only risk factor associated with a lack of humoral response, with an OR of 1.04 (95% CI 1.10–2.08), whereas previous SARS-CoV-2 infection was a protective factor (OR 0.19; 95% CI 0.04–0.98).

Table 3 shows the logistic regression analysis results for the factors related to cellular unresponsiveness. Hypogammaglobulinemia was also associated with a lack of cellular response (OR 8.5 95% CI (2.1–42.3)) and prior SARS-CoV-2 infection was a protective factor (OR 0.395% CI (0.02–0.9)).

An additional study regarding the cellular immunity of non-responders was performed. Thirteen SOT patients presented a non-reactive ELISpot test result after four doses of the mRNA-1273 vaccine. In six of them (four HTR and two LTR), the S-ELISpot switched to positive 6 months after the fourth vaccine dose, with no evidence of symptomatic SARS-CoV-2 infection or an extra booster dose during that 6-month follow-up.

In the univariate analysis (Table 4), SOT recipients with previous symptomatic SARS-CoV-2 infection presented higher rates of vaccine response (*p* = 0.003), and older patients presented a trend towards less vaccine responsiveness without reaching statistical significance (*p* = 0.06). However, when evaluating the humoral and cellular responses concurrently, no risk factor related to immune vaccine unresponsivenessne was identified.

### 3.4. SARS-CoV-2 Infection

Of the entire patient cohort, 22 (18%) were diagnosed with symptomatic SARS-CoV-2 infection. Among them, four (3.3%) presented severe infection requiring hospitalization, and one of them died.

SOT recipients with prior symptomatic SARS-CoV-2 infection had a stronger immune response, which enhanced the protective immune response, as was reflected in the logistic regression analyses. The humoral response (Table 2) had an OR of 0.19 (95% CI 0.04–0.98), while the cellular response (Table 3) had an OR of 0.2 (95% CI 0.04–0.09). Among vaccinated SOT recipients who had a symptomatic SARS-CoV-2 infection and never developed an immune response, 66% had severe infection that required hospitalization, and which led to death in one of them.

### 3.5. Safety

Only minor local adverse effects were noticed, with no noteworthy laboratory abnormalities or de novo DSA development. No episodes of rejection were registered during the initial follow-up period. These findings are consistent with those in the existing literature on solid organ transplant recipients [13].

## 4. Discussion

The most important finding of the present study was that mRNA SARS-CoV-2 vaccines showed a 97% global immune responsiveness in solid organ transplant recipients (antibody response 93%, cellular response 94%) with a reassuring safety profile. Importantly, hypogammaglobulinemia was associated with lower immune response, and previous symptomatic SARS-CoV-2 infection increased humoral and cellular protective response.

The high percentage of immunogenicity seen in the study is in some ways novel, but is consistent with recent literature describing better the responses to four vaccines compared with fewer SARS-CoV-2 vaccine doses [11,13]. These studies also described differences depending on the transplanted organ and immunosuppression regimen, finding higher responses in liver recipients reaching almost 96% of humoral response, and lower responses in kidney, lung, and heart [11]. One explanation of these optimistic results could be the use of the mRNA-1273 vaccine instead of others SARS-CoV-2 vaccines related to worse rates of immunogenicity [15]. Moreover, liver transplant recipients displayed better humoral responses after the second vaccine doses, and better cellular responses after each vaccine dose compared to other organs [16]. Vaccine response has been associated with time elapsed since transplant, with a higher risk of vaccine unresponsiveness during the first months after transplantation [17], and in the present study, the percentage of included patients that underwent transplantation in the previous year was low. Therefore, these optimistic results might be magnified, and the “real” percentage of immunogenicity could be lower than that reported.

Hypogammaglobulinemia was related to a lack of vaccine response, both humoral and cellular, as is consistent with the findings of recent studies [7,18,19]. This condition is common in SOT recipients [20,21,22] and often related to intense immunosuppressive therapies, especially in heart transplants, and it may be related to mycophenolate mophetil use, with an impact on B and T cell function. This condition has been related to an increased risk of opportunistic infections, particularly in the first 6 months post-transplantation. Accordingly, some experts have recommend the use of intravenous immunoglobulin during that period to minimize the risk of infection. Additionally, the immunogenicity of patients treated with B-cell-depleting therapies, such as rituximab, is impaired [23,24]. Other variables associated with vaccine unresponsiveness were lymphopenia and receiving high doses of mycophenolic acid [7,25,26]; it could be considered that these two variables are predominantly present in the early posttransplant period, and consequently, this association could not be established in the study. No other differences in baseline or treatment characteristics were found to be related to unresponsiveness.

Another important finding is that symptomatic SARS-CoV-2 infection improved immunogenicity. Prior studies performed in non-immunosuppressed populations proved that humoral immune response was boosted with SARS-CoV-2 infection, showing broader immunity compared to vaccination alone [27,28]. Furthermore, this phenomenon has recently also been demonstrated in SOT recipients [10,13,24,25,29,30]. Even though some studies found that infection-acquired immunity was greater than vaccine-acquired immunity, others found that both types of immunity are equivalent. Further studies performed on SOT recipients are needed to establish this association.

Of special concern is that eight SOT recipients presented an absence of antibodies and cells after four doses of the vaccine. Identifying these patients is crucial, since their risk of severe infection is higher, and they need advice on contact precautions and close contacts’ vaccination. Additional or delayed vaccine doses might help in boosting humoral and cellular responses to SARS-CoV-2 vaccines, and a heterologous strategy using both mRNA and viral vector vaccines could even enhance immunity [31].

Furthermore, exclusively local adverse events were documented, with no episodes of rejection or de novo DSA, as previously reported in the literature [32,33].

The limitations of the study could include the absence of a non-transplant control group, and the small size of the cohort. Moreover, baseline cellular responses were not evaluated, and some asymptomatic infections could have gone unnoticed; therefore, the incidence of SARS-CoV-2 infection could be underestimated. Furthermore, due to the limited patient follow-up, we were not able to describe the length of immune protection or its long-term behavior. Finally, as described before, only a few patients in the early post-transplant period were included, who were known to be at high risk of vaccine unresponsiveness due to high immunosuppressive load, thus the high vaccine response rates may be overestimated.

## 5. Conclusions

The mRNA-1273 SARS-CoV-2 vaccine demonstrated a robust 97% immunological responsiveness in our SOT recipients after combining antibody and cell responses. Hypogammaglobulinemia was related to a reduced vaccine response, whereas previous symptomatic SARS-CoV-2 infection was related to increased immune response.

## Figures and Tables

**Figure 1 vaccines-11-01845-f001:**
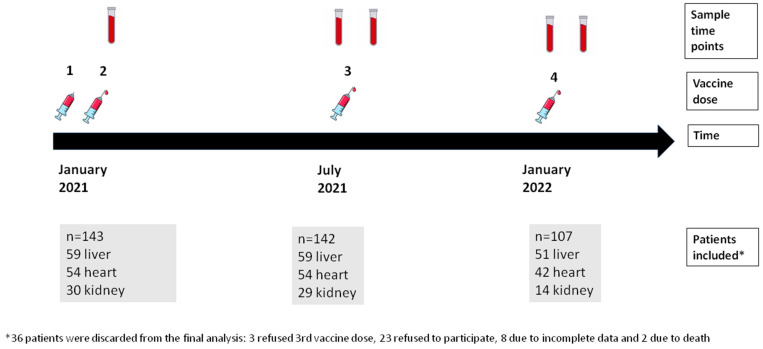
Study design showing sample time points, vaccine doses, and number of patients included in each sample time point.

**Figure 2 vaccines-11-01845-f002:**
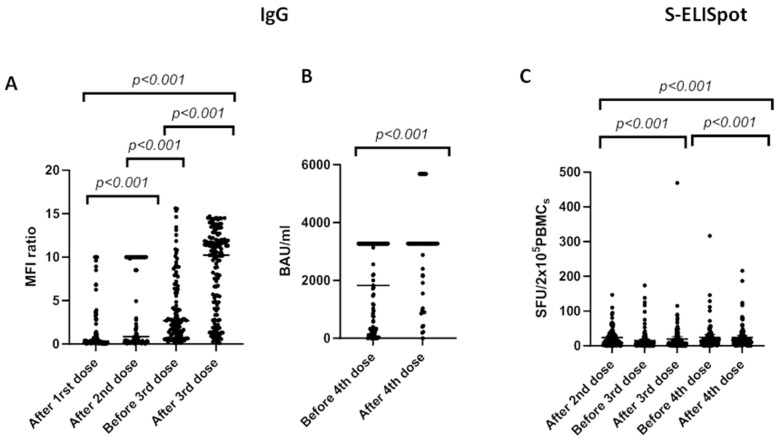
Changes in IgG concentration (**A**,**B**) and S-ELISpot (**C**) before and after different vaccine doses of mRNA-1273 SARS-CoV-2 vaccine. We used the Mann–Whitney test to compare means between groups and the Wilcoxon signed-rank test to compared samples at different time-points within the same group. Bars identify medians. MFI: median fluorescent intensities. BAU: binding antibody units. SFU: spot-forming units.

**Table 1 vaccines-11-01845-t001:** Baseline characteristics categorized by organ transplantation.

	Liver Recipients n = 59	Heart Recipients n = 54	Kidney Recipients n = 30	*p*-Value
Age (years), median	62 (13.5)	59 (14.3)	64 (12.8)	0.31
Sex (female)	28.8%	35.2%	40%	0.76
Hypertension	61%	26%	100%	<0.001
Diabetes	37.3%	26%	30%	0.42
BMI, median (IQR)	26.4 (17–42)	24.7 (18–36)	26.8 (18–39)	0.03
Dyslipidemia	35.6%	59%	86.7%	<0.001
HIV infection	3.4%	0	0	0.45
Median time from transplantation, years (IQR)	3.86 (0–27)	7.02 (0.6–26)	4.8 (0.6–42)	0.23
First year post transplantation	25.4%	7.4%	10%	0.02
Prior transplantation	1.7%	5.6%	10%	0.22
Acute rejection (last year)	12%	0%	3%	0.02
Induction (last 12 months)				0.6
ATG	7.5%	7.4%	13%	
Basiliximab	7%	0%	0%	
Rituximab	0%	0%	0%	
Immunosupressive regimen				<0.001
Quadruple/Triple therapy	17%	80%	63%	
Bitherapy	37.3%	20%	34%	
Monotherapy	44%	0%	3%	
No therapy	2%	0%	0%	
Type of immunosupressive drug				<0.001
Calcineurin inhibitors	89%	100%	100%	
Mycophenolate	29%	76%	60%	
Prednisone	29%	83.3%	80%	
mTOR inhibitors	20%	20.4%	20%	
Mycophenolic acid ≥ 1500 mg/day	10%	22%	0%	<0.001
Lymphopenia (<1000/mm^3^) (%yes)	8%	15%	13%	0.6
Hypogammaglobulinemia (<6.8 g/L IgG)	13.6%	13%	10%	0.7
Chronic kidney disease (GFR < 60 mL/min/1.73 m^2^)	33%	50%	17%	0.07

**Table 2 vaccines-11-01845-t002:** Factors with association to vaccine unresponsiveness (no SARS-CoV-2 IgG/IgM antibodies) 4–6 weeks after the last (4th) shot of the vaccine.

	Univariate Analysis			Multivariate Analysis
	Humoral Response n = 101	Absense of Humoral Response n = 7	*p*-Value	OR; 95% CI, *p* Value
Age (median, SD)	61 (SD12)	66 (SD6)	0.03	0.86; 0.72–1.02, 0.08
Sex (female)	33 (32%)	2 (28%)	1	
Hypogammaglobulinemia	10 (10%)	2 (28%)	0.05	1.04; 1.10–2.08, 0.04
Hypertension	70 (69%)	5 (71%)	1	
Diabetes mellitus	33 (33%)	2 (29%)	1	
Lymphopenia (<1000/mm^3^)	4 (4%)	0	1	
Vaccination at first year post transplantation	6.6 (SD7)	7.6 (SD 5)	0.9	
Mycophenolic acid ≥ 1500 mg/day	10 (10%)	2 (28%)	0.12	
SARS-CoV-2 infection	13 (13%)	3 * (43%)	0.06	0.19; 0.04–0.98, 0.04
Acute rejection (last year)	7 (7%)	0	1	

* two-thirds presented severe infection.

**Table 3 vaccines-11-01845-t003:** Factors related to cellular unresponsiveness (negative S-ELISpot response).

	Univariate Analysis			Multivariate Analysis
	Presence of S-ELISpot Response n = 76	Absense of S-ELISpot Response n = 8	*p* Value	OR; 95% CI, *p* Value
Age (median, SD)	61 (SD12)	65 (SD18)	0.7	
Sex (female)	22 (29%)	2 (25%)	0.6	
Hypogammaglobulinemia	4 (5%)	6 (75%)	0.014	8.5 (2.1–42.3), 0.008
Hypertension	50 (66%)	6 (76%)	0.8	
Diabetes mellitus	27 (36%)	3 (37%)	1	
Lymphopenia (<1000/mm^3^)	4 (5%)	0	0.7	
Vaccination at first year post transplantation	13 (18%)	2 (25%)	1	
Mycophenolic acid ≥ 1500 mg/day	13 (18%)	2 * (25%)	1	
SARS-CoV-2 infection	6 (8%)	3 (38%)	0.06	0.3 (0.02–0.9), 0.03
Acute allograft rejection (last year)	7 (5%)	0	0.7	

* 1/2 presented severe infection.

**Table 4 vaccines-11-01845-t004:** Outcomes of a logistic regression analysis, which examined the factors linked to the absence of both cellular and humoral responses (lack of IgG and negative ELISpot) one month following the administration of the fourth dose of the mRNA vaccine.

	Univariate Analysis			Multivariate Analysis
	Vaccine Response n = 104	Absence of Vaccine Response n = 3	*p* Value	OR; 95% CI, *p* Value
Age (median, SD)	61 (SD12)	76 (SD7)	0.06	1.3 (0.7–1.06), 0.06
Sex (female)	34 (33%)	1 (33%)	1	
Hypogammaglobulinemia	11 (78%)	1 (33%)	0.1	
Hypertension	72 (69%)	2 (67%)	1	
Diabetes mellitus	33 (32%)	1 (33%)	1	
Lymphopenia (<1000/mm^3^)	4 (4%)	0	1	
Vaccination at first year post transplantation	18 (17%)	0	1	
Mycophenolic acid ≥ 1500 mg/day	11 (10%)	0	1	
SARS-CoV-2 infection	13 (12%)	3 * (100%)	0.003	
Acute allograft rejection (last year)	7 (6%)	0	1	

* Two-thirds presented severe infection.

## Data Availability

The data that support the findings of this study are available on reasonable request from the corresponding author (M.F. or M.B.).

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
