# Peer review of "Impact of SARS-CoV-2 Infection on Humoral and Cellular Immunity in a Cohort of Vaccinated Solid Organ Transplant Recipients"

_vaccines, 2023, doi:10.3390/vaccines11121845_

Round 1

Reviewer 1 Report

Comments and Suggestions for Authors

The manuscript describes the evaluation of humoral and cellular responses in solid organ transplant recipients to mRNA-1273 vaccination (primary vaccination plus two boosters) and the effect of SARS-CoV-2 infection on both immune responses.

Although the Authors present an important study (there are still a limited number of publications on this topic concerning this group of patients), the manuscript was not completely prepared with the requirements of the Vaccines and the standards for scientific papers published in a journal with an impact factor of 7.8. Therefore, in order for the manuscript to be published in Vaccine, the Authors should revise it.

I have a few comments that I hope will allow the manuscript to be published in the future:

1.      A very important note: in scientific publications, we usually avoid writing in the personal form. The generally accepted form is to construct sentences like something was done, something was analysed, etc. The authors have already used the word "we" 3 times in the Abstract. Another example, in the Introduction, the word "our" is used in line 52. Please edit the entire text to avoid, if possible, the personal form.

2.      Please check the entire text to remove double spaces. The Reviewer already found such situations at the beginning of the evaluation of the manuscript: lines 38 and 52.

3.      Please keep the logical sequence of providing information about the manufacturer.  For example, in the Abstract, the type of vaccine is first listed on line 25, while the manufacturer information is provided on line 30. Please provide the manufacturer information in accordance with the Vaccines guidelines.

4.      Please provide a reference to the information from lines 48-50.

5.      Please "enable" the "do not split words" function in the Word menu and apply it to all text.

6.      In line 65, after indicating the author and co-authors (et al.), please begin the following word "evaluation" with a minor letter.

7.      Please explain for what purpose in the manuscript on the study of humoral and cellular responses, the Authors included data on detection of anti-HLA donor-specific HLA antibodies. In the opinion of the Reviewer, the results of these tests do not affect the studied parameters (assumed in the purpose of the study), and the inclusion of the results of the mentioned test, are only to show that the Authors have done an additional test, and thus through its description increased the volume of the manuscript.

8.      Materials and methods section: if the authors followed the information provided in the manufacturer's leaflet, detailed descriptions of the procedures used do not make sense. If the Authors used deviations, please indicate in the text only them.

9.      The Authors indicated in the Introduction that the results of the immune response (after two doses) have already been published (lines 52-55). So in this earlier publication, was the study group (patients) are not characterized? Are these the same patients? If these are the same patients (it seems that liver and heart recipients were previously described), the Reviewer does not see the validity of the point about the basic characteristics of the study group. Just specify only for "new" patients, and for the rest make a reference to the publication. In the Reviewer's opinion, the results previously published should not be detailed once again in the text.

10.  Please remove the scorers in the form of squares and dots from Table 1.

11.  Point 3.2: please replace : "2nd", "3rd", …. with the second and third, …, respectively.

12.  Please check the correctness of the descriptions of the Tables (some of them are bolded, e.g., Table 3, Table 4, on the other hand, is described incorrectly).

13.  Please check the correctness of the notation of references (for example, references 11 and 12 on line 132 and 13 on line 279 are in round brackets instead of square ones). Please check the entire text, as the Reviewer only gives examples.

14.  The Reviewer does not see information regarding detection of anti-HLA donor-specific HLA antibodies in the Discussion, all the more pointing to the earlier comment (Review Item 7).

In the Reviewer's opinion, the entire text needs improvement in the description style of all sections.

Comments on the Quality of English Language

In the Reviewer's opinion, the entire text needs improvement in the description style of all sections.

Author Response

Barcelona, December 2023

ID: vaccines-2697487

Dear reviewers,

Thank you very much for your review of our manuscript entitled “IMPACT OF SARS-COV-2 INFECTION ON HUMORAL AND CELLULAR IMMUNITY IN A COHORT OF VACCINATED SOLID ORGAN TRANSPLANT RECIPIENTS”. We truly think that your review has helped us to improve the article. According to your indications we provide a point-by-point response to your comments.

We look forward to hearing from you,

Sincerely,

Marta Bodro

Infectious Diseases Department

Hospital Clínic

08036 Barcelona

Barcelona, Spain

martabodro@clinic.cat

REVIEWER 1

The manuscript describes the evaluation of humoral and cellular responses in solid organ transplant recipients to mRNA-1273 vaccination (primary vaccination plus two boosters) and the effect of SARS-CoV-2 infection on both immune responses.

Although the Authors present an important study (there are still a limited number of publications on this topic concerning this group of patients), the manuscript was not completely prepared with the requirements of the Vaccines and the standards for scientific papers published in a journal with an impact factor of 7.8. Therefore, in order for the manuscript to be published in Vaccine, the Authors should revise it.

I have a few comments that I hope will allow the manuscript to be published in the future:

  1. A very important note: in scientific publications, we usually avoid writing in the personal form. The generally accepted form is to construct sentences like something was done, something was analysed, etc. The authors have already used the word "we" 3 times in the Abstract. Another example, in the Introduction, the word "our" is used in line 52. Please edit the entire textto avoid, if possible, the personal form.

Response: According to your suggestion we changed all sentences avoiding the personal form.

  1. Please check the entire text to remove double spaces. The Reviewer already found such situations at the beginning of the evaluation of the manuscript: lines 38 and 52.

R: We changed that.

  1. Please keep the logical sequence of providing information about the manufacturer.  For example, in the Abstract, the type of vaccine is first listed on line 25, while the manufacturer information is provided on line 30. Please provide the manufacturer information in accordance with the Vaccinesguidelines.

R: We changed that according to Vaccines guidelines.

  1. Please provide a reference to the information from lines 48-50.

R: We included two new references.

  1. Please "enable" the "do not split words" function in the Word menu and apply it to all text.

R: We corrected all text.

  1. In line 65, after indicating the author and co-authors (et al.), please begin the following word "evaluation" with a minor letter.

R: We corrected it.

  1. Please explain for what purpose in the manuscript on the study of humoral and cellular responses, the Authors included data on detection of anti-HLA donor-specific HLA antibodies. In the opinion of the Reviewer, the results of these tests do not affect the studied parameters (assumed in the purpose of the study), and the inclusion of the results of the mentioned test, are only to show that the Authors have done an additional test, and thus through its description increased the volume of the manuscript.

R: Another secondary objective of the study was to study the safety of the vaccine in SOT recipients (randomized trials did not include transplant recipients) and previous studies performed in transplant recipients found anti-HLA antibodies post vaccinations. We included this in the abstract section and discussed in the discussion section. 

  1. Materials and methods section: if the authors followed the information provided in the manufacturer's leaflet, detailed descriptions of the procedures used do not make sense. If the Authors used deviations, please indicate in the text only them.

R: We excluded all the information related to procedures.

  1. The Authors indicated in the Introduction that the results of the immune response (after two doses) have already been published (lines 52-55). So in this earlier publication, was the study group (patients) are not characterized? Are these the same patients? If these are the same patients (it seems that liver and heart recipients were previously described), the Reviewer does not see the validity of the point about the basic characteristics of the study group. Just specify only for "new" patients, and for the rest make a reference to the publication. In the Reviewer's opinion, the results previously published should not be detailed once again in the text.

R: Our previous study included only few patients of the present one. Moreover, kidney recipients were not included in the first study. We specified that.

  1. Please remove the scorers in the form of squares and dots from Table 1.

R: We removed that.

  1. Point 3.2: please replace : "2nd", "3rd", …. with the second and third, …, respectively.

R: We changed that.

  1. Please check the correctness of the descriptions of the Tables (some of them are bolded, e.g., Table 3, Table 4, on the other hand, is described incorrectly).

R: We corrected that.

  1. Please check the correctness of the notation of references (for example, references 11 and 12 on line 132 and 13 on line 279 are in round brackets instead of square ones). Please check the entire text, as the Reviewer only gives examples.

R: We corrected that.

  1. The Reviewer does not see information regarding detection of anti-HLA donor-specific HLA antibodies in the Discussion, all the more pointing to the earlier comment (Review Item 7).

R: We included information regarding detection of anti-HLA donor-specific HLA antibodies in the Discussion: “Furthermore, only local adverse events were reported with no episodes of rejection or de novo DSA in the immediate follow-up, in accordance with previous studies performed in SOT recipients”.

In the Reviewer's opinion, the entire text needs improvement in the description style of all sections.

REVIEWER 2

The authors condudcted prospective study to determine humoral and cellular immunity response to 4 doses of mRNA vaccine in donors of liver heart and kidney transplant recipients. They concluded that vaccination is effective in inducing humoral and cekllular imunity in most recipients and its safe and well tolerated. Liver transplant recipients had better response than kidney and heart recipeints. They resulsts were comparable but in some extent more optimistic than results of other outhors. The efficacy of vacination in organ recipients has been reported in other studies however authors of this study demonstrated humaroal and cellular response in higher proportion of vaccinated patients. The authors included recipients of mRNA-1273 vaccine.

Minor revisions: 

- Vaccine mRNA-1273 demonstrated higher efficacy than other mRNA vaccines in some studies. Please discus if the use of thos vaccine coud lead to more optimistic results in your study. 

R: We changed this paragraph and improved it.

- In the dicusion, authors stated: "Vaccine response has been associated time elpased since transplant, with higher risk of vaccine unresponsiveness during the first months after transplantation. In these studies, the percentage of included patients that underwent transplantation in the previous year was low. Therefore, our optimistic results might be magnified..." Reader might be uncertain which studies do authors refer to. Consider better formulation.  

R: We improved that paragraph.

- State the originality of the results within the first paragraphs of discusion. 

R: We changed that.

Reviewer 2 Report

Comments and Suggestions for Authors

The authors condudcted prospective study to determine humoral and cellular immunity response to 4 doses of mRNA vaccine in donors of liver heart and kidney transplant recipients. They concluded that vaccination is effective in inducing humoral and cekllular imunity in most recipients and its safe and well tolerated. Liver transplant recipients had better response than kidney and heart recipeints. They resulsts were comparable but in some extent more optimistic than results of other outhors. The efficacy of vacination in organ recipients has been reported in other studies however authors of this study demonstrated humaroal and cellular response in higher proportion of vaccinated patients. The authors included recipients of mRNA-1273 vaccine.

Minor revisions: 

- Vaccine mRNA-1273 demonstrated higher efficacy than other mRNA vaccines in some studies. Please discus if the use of thos vaccine coud lead to more optimistic results in your study. 

- In the dicusion, authors stated: "Vaccine response has been associated time elpased since transplant, with higher risk of vaccine unresponsiveness during the first months after transplantation. In these studies, the percentage of included patients that underwent transplantation in the previous year was low. Therefore, our optimistic results might be magnified..." Reader might be uncertain which studies do authors refer to. Consider better formulation.  

- State the originality of the results within the first paragraphs of discusion. 

Author Response

(The authors gave the same response as above.)

Round 2

Reviewer 1 Report

Comments and Suggestions for Authors

The authors have mostly responded to previous comments.

I have a few more minor comments.

1.      Line 49: Reference 3 is posted twice. Please remove one.

2.      Lines 49-51: Please add two references; documents of the European and American transplant societies.

3.      The authors did not address the comment: Please "enable" the "do not split words" function in the Word menu and apply it to all text. Lines: 53, 55, 56, 57, 59, 61, etc. Please check the full text.

4.      Line 88: Before the authors use the abbreviation "S" protein, the full name, spike (S) protein, should be given.

5.      Line 263: The reference should be in square brackets. Please remove the dot before the bracket.

6.      A general comment on the entire text: Please make spaces before references.

7.      In characterizing the patients, the authors determined their body mass index (BMI). The reviewer did not find a comment in the Discussion on whether the patient's BMI impacted the immune response.

Author Response

Barcelona, December 2023

ID: vaccines-2697487

Dear reviewers,

Thank you very much for your review of our manuscript entitled “IMPACT OF SARS-COV-2 INFECTION ON HUMORAL AND CELLULAR IMMUNITY IN A COHORT OF VACCINATED SOLID ORGAN TRANSPLANT RECIPIENTS”. We truly think that your review has helped us to improve the article. According to your indications, we provide a point-by-point response to your comments.

We look forward to hearing from you,

Sincerely,

Marta Bodro

Infectious Diseases Department

Hospital Clínic

08036 Barcelona

Barcelona, Spain

martabodro@clinic.cat

  1. Please confirm the order of author name (Marta Bodro is in the first in
    the system but the last in the Word document).

R: Marta Bodro is the last author of the manuscript (together with Marta Farrero) whereas Bernardo Ayala-Borges is the first. 

  1. Please confirm the newly added "Conclusion" section.

R: We confirmed the conclusion section

  1. The similarity is still higher than our requirement, which cannot be 
    accept for further process. We attached the latest iThenticate report and a Word document with highlighted sentences, please revise these sentences

R: We have modified the majority of sentences with high similarity. Please be aware that most of the similarity is due to the methods section, which is the same as in our last paper, so probably that’s where the similarity comes from. Nevertheless, we rephrased it as much as we could without changing the meaning or the techniques used, hopefully, it will be different enough in the current version.

REVIEWER 1 ROUND 2

The authors have mostly responded to previous comments.

I have a few more minor comments.

1.Line 49: Reference 3 is posted twice. Please remove one.

R: Thanks for noticing, we have amended this mistake.

  1. Lines 49-51: Please add two references; documents of the European and American transplant societies.

R: We included the references.

  1. The authors did not address the comment: Please "enable" the "do not split words" function in the Word menu and apply it to all text. Lines: 53, 55, 56, 57, 59, 61, etc. Please check the full text.

R: We corrected all text.

4.Line 88: Before the authors use the abbreviation "S" protein, the full name, spike (S) protein, should be given.

R: We changed that.

5.Line 263: The reference should be in square brackets. Please remove the dot before the bracket.

R: We changed that.

6.A general comment on the entire text: Please make spaces before references.

R: We corrected all text.

  1. In characterizing the patients, the authors determined their body mass index (BMI). The reviewer did not find a comment in the Discussion on whether the patient's BMI impacted the immune response.

R: The body mass index was included in the baseline characteristics of the patients. However, in the statistical analysis, there were no significant differences in terms of immune response related to BMI nor other baseline characteristics, so we did not include them in the discussion. We added a sentence to clarify that no other factors were found relevant to predict immune response.

See line 353: “No other differences in baseline or treatment characteristics were found to be related to unresponsiveness.”